# Surveys on Pet-Reptile-Associated Multi-Drug-Resistant *Salmonella* spp. in the Timișoara Metropolitan Region—Western Romania

**DOI:** 10.3390/antibiotics12071203

**Published:** 2023-07-19

**Authors:** János Dégi, Viorel Herman, Isidora Radulov, Florica Morariu, Tiana Florea, Kálmán Imre

**Affiliations:** 1Department of Infectious Diseases and Preventive Medicine, Faculty of Veterinary Medicine, University of Life Sciences “King Mihai I”, 300645 Timisoara, Romania; viorelherman@usvt.ro; 2Faculty of Agriculture, University of Life Sciences “King Mihai I”, 300645 Timisoara, Romania; isidora.radulov@usvt.ro; 3Department of Animal Production Engineering, Faculty of Bioengineering of Animal Recourses, University of Life Sciences “King Mihai I”, 300645 Timisoara, Romania; 4Department of Dermatology, Faculty of Veterinary Medicine, University of Life Sciences “King Mihai I”, 300645 Timisoara, Romania; sujic.tijana@yahoo.com; 5Department of Animal Production and Veterinary Public Health, Faculty of Veterinary Medicine, University of Life Sciences “King Mihai I”, 300645 Timisoara, Romania; kalmanimre@usvt.ro

**Keywords:** reptiles, *Salmonella*, antibiotic resistance, public health

## Abstract

The number of reptiles owned as pets has risen worldwide. Additionally, urban expansion has resulted in more significant human encroachment and interactions with the habitats of captive reptiles. Between May and October 2022, 48 reptiles from pet shops and 69 from households were sampled in the Timișoara metropolitan area (western Romania). Three different sample types were collected from each reptile: oral cavity, skin, and cloacal swabs. *Salmonella* identification was based on ISO 6579-1:2017 (Annex D), a molecular testing method (*invA* gene target), and strains were serotyped in accordance with the Kauffman–White–Le-Minor technique; the antibiotic susceptibility was assessed according to Decision 2013/652. This study showed that 43.28% of the pet reptiles examined from households and pet shops carried *Salmonella* spp. All of the strains isolated presented resistance to at least one antibiotic, and 79.32% (23/29) were multi-drug-resistant strains, with the most frequently observed resistances being to gentamicin, nitrofurantion, tobramycin, and trimethoprim–sulfamethoxazole. The findings of the study undertaken by our team reveal that reptile multi-drug-resistant *Salmonella* is present. Considering this aspect, the most effective way of preventing multi-drug-resistant *Salmonella* infections requires stringent hygiene control in reptile pet shops as well as ensuring proper animal handling once the animals leave the pet shop and are introduced into households.

## 1. Introduction

*Salmonella* species are widely recognized as one of the most significant zoonotic diseases. Six percent of human Salmonellosis cases are acquired after interactions with reptiles, and the intimate contact between reptiles and their owners creates favorable conditions for the transfer of zoonotic pathogen infections [1,2,3]. In addition, *Salmonella* strains discovered from pet reptiles have been linked to antibiotic resistance, one of the most serious health issues of the twenty-first century that may have therapeutic repercussions for reptile owners and breeders [1,4,5,6]. 

*Salmonella* causes serious illness in people and animals [3,7]. A wide range of *Salmonella* species are ubiquitous in various environments, harboring the capacity to induce infections in both humans and animals [4,8]. *Salmonella* can occasionally cause more severe illnesses such as bacteremia (with or without metastatic disease), skin and bone infections, UTIs, meningitis, osteomyelitis, and splenic abscesses [7,8,9,10,11,12,13]. *Salmonella* is most associated with contaminated food and is responsible for 80 million annual cases of Salmonellosis [14].

Captive reptiles are a common source of *Salmonella*, and cases of Salmonellosis in households with reptiles are rising [15,16].

Therefore, it is imperative that all reptiles maintained as pets be viewed as a potential reservoir for human infection, and one must acknowledge their contribution to the increased risk of therapeutic failure in Salmonellosis cases with possibly fatal outcomes in both human and veterinary medicine [3,4]. Despite the rise in the popularity of snakes and lizards as pets worldwide, the true prevalence of *Salmonella* in the pet trade is likely under-reported due to the lack of studies examining this topic in depth. As the market for exotic pets continues to grow, there is a risk that diseases will spread worldwide as reptiles are imported from other countries [17,18,19]. Many species of reptiles have become increasingly popular pet animals during the past 20 years, even though species are protected by the Convention on International Trade in Endangered Species (CITES) [20]. The increasing popularity of reptiles as household pets has prompted worries about spreading *Salmonella* bacteria to humans [21,22].

Contaminated food and water serve as sources for disease in mammals. *Salmonella* infections rely on the capacity of bacteria to thrive in harsh living conditions, such as those in the digestive tract, prior to entering the intestinal epithelium and proceeding to colonize the mesenteric lymph nodes and internal organs, subsequently leading to systemic infections [7,23,24].

The antibiotic resistance of bacteria to antimicrobials is currently a primary concern in both human and veterinary medicine. For this reason, epidemiological studies in domestic and wild animals should be performed on a regular basis [25]. Resistant pathogens, including *Salmonella enterica*, should be paid particular attention, as these bacteria are very well adapted to different hosts, carry different genes encoding for both virulence and antimicrobial resistance, and are currently among the most common infectious agents isolated from humans with food-borne infections. Although many countries have strict antimicrobial use restrictions, enforcement is sometimes lax, resulting in the indiscriminate use of antimicrobials. This abuse has aided in the evolution of multi-drug-resistant (MDR) bacterial strains [25,26], and infections with these bacteria are on the rise. *Salmonella* strains that are resistant to antibiotics have been discovered in reptiles all over the world [5,27]. When humans become infected with resistant *Salmonella* strains, therapy can be challenging, increasing the probability of treatment failure and, in extreme cases, death [27,28]. As a result, understanding the antibiotic resistance patterns of *Salmonella* spp. infections in reptiles is critical.

The present study focuses on the evaluation of multi-drug-resistant *Salmonella* carriage by pet reptiles in pet stores and households, as well as its role in the transmission of antimicrobial resistance, to inform owners about potential risk factors.

## 2. Results

From all of the samples collected during this study, 25.46% (41/161) tested positive for *Salmonella* according to the conventional and molecular detection of the *invA* gene (~284 bp). 

*Salmonella* carriage was substantially linked by sample type (*p* = 0.0386; 95% CI = 1.0366% to 34.9828%) with greater positive samples from cloacae (35.29%, 18/51) than from the skin (23.88%, 16/67) and oral cavities (16.27%, 7/43) (Table 1). 

*Salmonella* spp. were detected in 43.28% (29/67) of the individuals sampled (Table 2), without significant differences between snakes (77.76%, 7/9) and lizards (44.18%, 19/43) compared to chelonians (20.0%, 3/15) (*p* = 0.065; 95% CI = −4.2750% to 40.8743%).

This study selected reptiles from private owners (56.71%, 38/67) and pet shops (43.28%, 29/67) in the Timisoara metropolitan region (western Romania). *Salmonella* isolation was higher in pet shop reptiles (68.96%, 20/29) than in household pets (23.68%, 9/38) (*p* = 0.002; 95% CI = 21.3630% to 62.7043%).

Moreover, the number of reptiles cohabiting the same terrarium was known for 49 of the 67 reptiles analyzed; 28 came from pet shops and 21 from households. In pet shops, no significant differences were found between the number of reptiles in the same terrarium and *Salmonella* shedding (*p* = 0.304; 95% CI = −13.0774% to 45.9947%). Thus, 88.23% of the reptiles that cohabited terrariums with two or more reptiles were positive for *Salmonella* (15/17), while 72.73% of reptiles that inhabited terrariums alone were positive for the bacterium (8/11). In contrast, for private owners’ reptiles, no significant differences were observed between reptiles that cohabited terrariums with two or more reptiles or alone for *Salmonella* shedding (*p* = 0.1736; 95% CI = −10.2386% to 59.0514%): 16.67% (2/12) and 44.45% (4/9), respectively (Table 2). 

*Salmonella* carriage and diets were not found to be tightly associated (*p* = 0.4809; 95% CI = −21.3475% to 38.9895%), with reptiles that were fed animal-originated food surpassing reptiles that were fed vegetable-originated food or processed food, in terms of frequency: 46.34% or 19/41 for the first category; 41.17% or 7/17 for the second category; and 33.33% or 3/9 for the third category (Table 2).

From the total of 67 collected samples, 61.20% (41/67) came from carnivorous reptiles, 25.37% (17/67) from herbivorous reptiles, and 13.43% (9/67) from omnivores (Table 3). The distribution of *Salmonella*-positive samples (Table 3) revealed a significant difference between the positive samples from carnivorous reptiles (21/29; 72.41%) and the other two categories of reptiles, omnivorous (5/29; 17.24%) and herbivorous (3/29; 10.34%). The value of *p* confirmed the significant difference (*p* = 0.0395; 95% CI = 3.8510% to 79.4251%).

From the 29 strains selected for serotyping, 23 were viable after culture and were serotyped. All of the *Salmonella* isolates were classified as *Salmonella enterica*. The most represented subspecies were *S. enterica* (47.82%, 11/23), *S. diarizonae* (21.73%, 5/23), *S. houtenae* (17.39%, 4/23), and *S. arizonae* (13.04%, 3/23). 

Twenty-three different serovars of *S. enterica* subspecies were identified (Table 4). 

Out of the 41 *Salmonella* strains isolated, 32 were viable after culture and were included in the antimicrobial susceptibility study. All of the strains analyzed resisted at least one of the sixteen antibiotics tested (n = 32/32). The highest percentages of antibiotic resistance were found in aminoglycosides: GM/84.37%, n = 27; TM/56.25%, n = 18; AN/28.12%, n = 9; and then diaminopyrimidine with sulfonamide: SXT/71.87%, n = 23; nitrofuran derivative: FT/34.38%, n = 11; and penicillin: AM/21.87%, n = 11 (*p* = 0.0003; 95% CI = 3.8510% to 79.4251%). The antimicrobial resistances of the different *Salmonella enterica* strains are summarized in Table 5.

Table 6 summarizes the antimicrobial resistance patterns of the various *Salmonella enterica* serovars. From the total of 23 *Salmonella* serovars tested, 60.86% (14/23) also exhibited resistance to three or more antibiotics, being included in strains with multiple resistance to antibiotics. There were 12 distinct resistance patterns in total (Table 6). 

## 3. Discussion

Reptiles being an important link in the carriage of *Salmonella* spp. worldwide is a commonly known fact. This carrier status of reptiles is reflected in the health hazards that they present for humans, especially children [1,3,5,29]. Despite this, there are no regulations in place when it comes to the implications of reptile pet shops in the spreading of MDR *Salmonella* strains. The findings from this study show that the number of *Salmonella* strains isolated from reptiles acquired from pet shops was double that compared to reptiles originating from private households (67 vs. 33%). The poor hygienic management of terrariums could be the cause behind this, mainly in pet shops where staff are usually busy and lack the necessary time to properly tend to cleanliness. Thus, ensuring proper cleaning and disinfection is difficult [30,31]. This situation encourages the persistence of MDR strains within shops, affecting consecutive batches of reptiles. Cohabitation with individuals that are of different ages or origins induces stress which, in turn, affects the reptiles’ resistance towards bacterial infection, enhances shedding within the terrarium, and facilitates reptile-to-reptile transmission [1]. Conversely, reptiles from private owners are exposed to better hygiene practices and less stressful environments, leading to lower *Salmonella* shedding [1].

In reptiles, *Salmonella* is spread by the fecal–oral route with the asymptomatic natural colonization of the enteric tract. In this study, the cloacal swabs collected were more sensitive to *Salmonella* isolation than other samples, such as those from skin or oral cavities. However, it is essential to highlight this because *Salmonella* is excreted through feces and could contaminate the reptile’s skin, oral cavity, and the environment, being a source of infection for humans who handle the reptile or who are exposed to the reptile’s habitat. Moreover, the *Salmonella* serovars most frequently detected in this study have been cited previously in reptile studies [1,3] and outbreaks [1]. The subspecies *houtenae*, *diarizonae*, *salamae*, and *arizonae* have been reported as species that are harbored mainly by cold-blooded animals which act as main reservoirs [1,10,32]. This study showed higher *Salmonella* prevalence among snakes and lizards, compared to the higher prevalence found in chelonians in previous research [1,3,15,33]. Particular attention has recently been oriented towards snakes and lizards due to increased interactions between them and humans in domestic environments [1,3,34]. In this sense, it is important to highlight that these reptiles are mainly fed with animal food, representing an important source of *Salmonella* [15,21,29,35,36]. Previous studies in the United Kingdom reported the important role of commercial feeder rodents in bacterial transmission among reptiles and even their owners [1,37]. Human *Salmonella* outbreaks have been correlated with *Salmonella*-contaminated feeder rodents and the cross-contamination of kitchens that serve as food preservation and preparation locations (keeping rodents in the freezer and thawing them in the microwave, for example) [37,38]. In this context, to avoid *Salmonella* infection in reptiles, the control of food products of animal origin must be mandatory for food suppliers [39]. On the other hand, due to their popularity as pets for children, special attention must be given to chelonians. Consequently, several countries, the US included, have implemented strict bans to curtail chelonian-associated salmonellosis; however, in Europe, there are few regulations to control its prevalence [40]. 

The present study isolated *Salmonella* from 18.5% of the screened chelonians. Seasonal effects, such as hibernation or the season of sampling, have been speculated by previous studies to explain the low isolation rate of *Salmonella* in chelonians compared to other reptiles [1,41,42,43]. Moreover, diet may also have an important role [42,43] because, as reported above, many chelonians are fed food that is of vegetable origin or that is processed, and not from animal origin, which is frequently related to *Salmonella* outbreaks [43].

In the current study, some isolated strains of *Salmonella* were found to have high levels of resistance to antibiotics used to treat infections in humans and pet reptiles, raising the possibility that humans could become infected with multi-drug-resistant *Salmonella* through contact with reptiles. Fluoroquinolones are an important class of antibiotics used to treat a variety of human and animal (including reptiles) infections, and they are especially effective against salmonellosis. To keep fluoroquinolones as effective as possible, they must be used with caution; antibiotic residues in food must be checked on a regular basis, and extensive monitoring for the formation of bacterial resistance in both animals and humans must be provided [44,45]. Carbapenems (ertapenem and imipenem), known as “last resort” antibiotics for use in cases where drug resistance monitoring is required, are required to establish any possible links between bacterial reservoirs and to limit the bidirectional transfer of encoding genes between *Salmonella* spp. and other commensal or pathogenic bacteria. *S. enterica* is a nosocomial infection in humans, and the frequency with which it is discovered to be resistant to both carbapenems is low when compared to other *Enterobacteriaceae* [8,46,47].

## 4. Materials and Methods

### 4.1. Sample Collection

Between May and October 2022, 29 reptiles from pet shops and 38 from households were sampled in the Timisoara metropolitan area (western Romania) and screened to estimate the prevalence of multi-drug-resistant *Salmonella* spp. Three different sample types were collected from each reptile: oral cavity, skin, and cloacal swabs. *Salmonella* identification was based on ISO 6579-1:2017 (Annex D), a molecular testing method (*invA* gene target), strains were serotyped using the Kauffman–White–Le-Minor technique, and the antibiotic susceptibility was assessed according to Decision 2013/652. All of the pet reptiles were presented for veterinary services at the University Veterinary Clinics of the Faculty of Veterinary Medicine, Timișoara, for a routine clinical examination. 

All sampling techniques were carried out in accordance with national standards and regulations. The collection of the specimens was performed with the consent of owners and breeders, according to the code of the Romanian Veterinary College (protocol numbers 34/1.12.2012), and the procedures followed those used by the University Veterinary Clinics of the Faculty of Veterinary Medicine, Timisoara. A standard procedure was established to recruit clinically healthy pet reptiles in the study. 

Previously, the owners and breeders were contacted by advertising the project through the University Veterinary Clinics of the Faculty of Veterinary Medicine Timisoara, Romania, and through private veterinary clinics and breeders in the Timișoara metropolitan region (western Romania). A total of 26 species were identified from the 67 sampled reptiles (Table 1). From these species, 7 were classified as chelonians (order *Chelonia* and *Testudines*), 12 as lizards (order *Squamata*), and 7 as snakes (order *Squamata* and *Serpentes*) (Table 1). According to the individuals sampled from each group, 64.17% (43/67), 22.38% (15/67), and 13.43% (9/67) were lizards, chelonians, and snakes, respectively (Table 7).

Whenever possible, samples from the oral cavity (n = 43), skin (n = 67), and cloaca (n = 51) were taken using sterile cotton swabs (ESwabs, Copan, Italy), according to standards CLSI M40-A2 (Quality Control of Microbiological Transport System) [48]. All of the individuals sampled were healthy, and none presented clinical symptoms such as diarrhea at the time of sampling. In addition, an epidemiological questionnaire was filled out. The questionnaire contained information about the species, diet, and the number of reptiles cohabiting the same terrarium. The diet was classified as food of animal origin (including live prey, fresh meat, and frozen meat), food of vegetable origin (including fruit and vegetables), and processed food (including commercially manufactured reptile food).

Moreover, the number of reptiles coexisting in the same terrarium was recorded as reptiles inhabiting terrariums alone or cohabitating with two or more reptiles. Sterile plastic containers were used to directly collect fecal samples from the cloaca or due to spontaneous emission. The harvested specimens were stored in refrigerated boxes and transported to the laboratory quickly. 

### 4.2. Bacterial Isolation

The *Salmonella* spp. were isolated using conventional methods and followed the protocols recommended by the Food and Drug Administration Agency (FDA) and Commission Regulation (EC) No. 2073/2005 [49], and according to the International Standard Organization standard, by ISO 6579-1:2017, Annex D [50], and they were serotyped using the Kauffman–White–Le-Minor technique. Antibiotic susceptibility was assessed according to Decision 2013/652. The samples were processed on the day of sampling in the Bacterial Disease’s Diagnostic Laboratory (B.6.a) of the Department of Infectious Diseases and Preventive Medicine of the Faculty of Veterinary Medicine, Timișoara. 

Briefly, within the first step, the collected samples were homogenized with 10 mL of selenite cystine broth (Merck, Bucharest, Romania) for 5 min, according to ISO 6579 [51]. The mixture was then incubated at 35 °C in an aerobic atmosphere for 24 h. Next, one milliliter from each pre-enriched sample after the incubation period was added to 10 mL of BD Rappaport Vassiliadis broth (BD Diagnostic Systems, Heidelberg, Germany) and mixed for 2 min, then subsequently incubated in the aerobic atmosphere for 24 h at 42 °C. After incubation, the tube content was stirred and inoculated on BD MacConkey agar II plates (BD Diagnostic Systems, Heidelberg, Germany) with a bacteriological inoculation loop. The inoculated BD MacConkey agar II plates were incubated at 35 °C in an aerobic atmosphere for 24 h. After this stage, presumptive *Salmonella* spp. colonies with characteristic morphologies on the BD BBL™ XLD agar (BD Diagnostic Systems, Heidelberg, Germany) media were tested to determine their biochemical characteristics, including xylose fermentation, lysine decarboxylation, and the production of hydrogen sulfide. The plates were incubated at 37 °C in an aerobic atmosphere for 24 h.

The *Salmonella* species were identified using the Vitek^®^ 2 Compact system (bioMérieux, Marcy l’Etoile, France) and the Vitek^®^ 2 GN–ID card, designed for the automated identification of the most clinically significant fermenting and nonfermenting Gram-negative bacilli [52].

### 4.3. Molecular Detection of invA Gene

All biochemically identified *Salmonella* strains were directly subjected to molecular analysis. Bacterial genomic DNA was isolated from the strains cultivated on selenite cystine broth (Merck, Bucharest, Romania) media using the QIAamp DNA Mini Kit (Qiagen partner, BioMarker Kft, Gödöllö, Hungary), according to the manufacturer’s specifications. The extracted DNA quantity and quality were determined using a UV–VIS Model UV-3100PC spectrophotometer (VWR International Kft., Debrecen, Hungary), measuring the absorbance at 260 nm. The strains were molecularly tested for the presence of the *Salmonella*-specific *invA* gene (~284 bp) using a conventional polymerase chain reaction, as was previously described by Kardy et al. [53]. The specific forward (5′ GTG AAA TTA TCG CCA CGT TCG GGC AA-3′) and reverse (5′ TCA TCG CAC CGT CAAAGG AAC C-3′) primers were used [44]. The PCR conditions consisted of an initial denaturation step at 95 °C for 5 min, followed by 32 cycles of denaturation at 95 °C for 1 min, annealing at 55 °C for 1 min, extension at 72 °C for 1 min, and a final extension at 72 °C for 10 min, using the Roche LightCycler 480-II PCR w (Roche, Diagnostic Division, Bucharest, Romania) thermocycler. All of the PCR amplicons were visualized on ethidium-bromide-stained 2.5% agarose gel under UV light using a UV Roth 254 nm, 4 × 6 W system (Roth Parter, Amex Import-Export, Bucharest, Romania). The strain *Salmonella enterica* serovar ATCC 13,076 was used as a positive control, and the negative control consisted of sterile deionized water.

### 4.4. Serotyping via Slide Agglutination (Kauffmann–White–Le-Minor Scheme)

The serotyping of the *Salmonella* isolates was achieved in a pure culture based on the evidence of somatic (O) and flagellar (H) antigens through reactions with specific antisera [51,54]. *Salmonella* O and *Salmonella* H antisera (SSI Diagnostica A/S, Hillerød, Denmark) were used according to the manufacturer’s recommendations.

### 4.5. Antimicrobial Susceptibility Testing

*Salmonella* antimicrobial susceptibility was tested in the 32 out of 41 *Salmonella* strains isolated that were viable after culture according to the Clinical and Laboratory Standards Institute (CLSI) guidelines [55].

Antimicrobial susceptibility testing of the isolated *Salmonella* strains was achieved with the Vitek 2^®^ automated equipment and the AST GN67 card (bioMérieux. Marcy l’Etoile, France). 

The tested antimicrobials were the following: amikacin (AN; MIC range 16–64 μg/mL), ampicillin/sulbactam (SAM; MIC range 8/4–32/16 μg/mL), cefazolin (CZ; MIC range 2–8 μg/mL), cefepime (FEP; MIC range 2–16 μg/mL), ceftazidime (CAZ; MIC range 4–16 μg/mL), ceftriaxone (CRO; MIC range 1–4 μg/mL), ciprofloxacin (CIP; MIC range 0.06–1 μg/mL), ertapenem (ETP; MIC range 0.5–2 μg/mL), gentamicin (GM; MIC range 4–16 μg/mL), imipenem (IPM; MIC range 1–4 μg/mL), levofloxacin (LEV; MIC range 0.12–2 μg/mL), nitrofurantoin (FT; MIC range 32–128 μg/mL), piperacillin/tazobactam (TZP; MIC range 16/4–128/4 μg/mL), tobramycin (TM; MIC range 4–16 μg/mL), and trimethoprim/sulfamethoxazole (SXT; MIC range 2/32–4/76 μg/mL). The system automatically processed the obtained results, and the isolates were categorized as susceptible, resistant, or intermediate. The isolates that were resistant to three or more antimicrobials were classified as multi-drug-resistant.

### 4.6. Statistical Analysis

To determine whether there was a correlation between categorical factors and the occurrence of Salmonella, AMR, and MDR, a generalized linear model was applied to the data (reptile species, habitat, sample type, diet, and number of reptiles cohabiting the same terrarium). If *Salmonella* was detected in the oral cavity, skin, or cloaca of a reptile, it was considered positive. For a statistically significant difference, the threshold was *p* ≤ 0.05. The results were analyzed using basic descriptive statistics. The 95 per cent confidence intervals (95% CI) for the *Salmonella enterica* strains isolated from reptiles were calculated and estimated to compare different factors. The obtained results were statistically interpreted using the SPSS statistical analysis software package, version 28.0.1.1, Chicago, USA. A nonparametric Pearson’s chi-squared (χ^2^) test was used to find any possible associations between the *Salmonella* infection status and the recorded epidemiological data. Differences were established as statistically significant when the *p*-value  ≤ 0.05.

## 5. Conclusions

There is evidence from this study that pet reptiles may be a source of *Salmonella* MDR for humans. The MDR problem in reptiles may begin in pet shops, where the presence of *Salmonella* is particularly high, and it appears to be connected to the origins of the reptiles’ food. Rising rates of human contact with reptiles may raise the risk of disease transmission. It was often observed that the patient did not understand the dangers of reptile ownership and did not follow standard precautions when handling the animals or cleaning their cages. This emphasizes the necessity for outreach efforts to educate the public about preventing salmonellosis from reptiles kept as pets. Against this background, the optimum prevention of *Salmonella* MDR infections implies the stringent sanitary surveillance of reptile shops and the proper hygienic manipulation of individuals in the household. However, it is essential to point out that the number of specimens enrolled is rather limited, and this may narrow the interpretation of our results to the Timisoara–western Romania metropolitan region. Further research is required to confirm our results on a larger study sample.

## Figures and Tables

**Table 1 antibiotics-12-01203-t001:** *Salmonella* spp. isolated by reptile sample type.

Reptiles’ Classification	Type of Samples	*Salmonella* spp. Carriage
Order *Squamata* (lizards)	Cloaca (n = 28)	9/28 (32.14%)
	Skin (n = 43)	8/43 (18.6%)
	Oral cavity (n = 23)	3/23 (13.04%)
Order Squamata/Serpentes *(snakes)*	Cloaca (n = 9)	5/9 (55.56%)
	Skin (n = 9)	4/9 (44.45%)
	Oral cavity (5)	2/5 (40.0%)
Order Chelonia/Testudines *(chelonians)*	Cloaca (n = 15)	4/15 (26.67%)
	Skin (n = 15)	4/15 (26.67%)
	Oral cavity (n = 15)	2/15 (13.34%)

**Table 2 antibiotics-12-01203-t002:** Distribution of *Salmonella* spp.-positive cases according to different factors.

Factor	Number of Positive-*Salmonella* Samples (%)
Type of reptiles	
Snakes (*n* = 9)	7 (77.76%)
Lizards (*n* = 43)	19 (44.18%)
Chelonians (*n* = 15)	3 (20.0%)
The keeping places	
Reptiles at private owners (*n* = 38)	9 (23.68%)
Reptiles at pet shops (*n* = 29)	20 (68.96%)
Type of cohabitation	
Reptiles cohabiting a terrarium—private owners (*n* = 21)	6 (12.24%)
Cohabitation with two or more reptiles (*n* = 12)	2 (16.67%)
Inhabiting terrariums alone (*n* = 9)	4 (44.45%)
Reptiles cohabiting a terrarium—pet shops (*n* = 28)	23 (46.94%)
Cohabitation with two or more reptiles (*n* = 17)	15 (88.23%)
Inhabiting terrariums alone (*n* = 11)	8 (72.73%)
Type of diet	
Reptiles fed with food of animal origin (*n* = 41): first category diet	19 (46.34%)
Reptiles fed with food of vegetable origin (*n* = 17): second category diet	7 (41.17%)
Reptiles fed with processed food for omnivorous animals (*n* = 9): third category diet	3 (33.33%)

**Table 3 antibiotics-12-01203-t003:** The distribution of Salmonella strains according to the reptile species and the type of diet.

Common Name of Reptiles	Type of Diet	Number of Reptiles Examined	Positive Samples for *Salmonella* spp. (%)
Western girdled lizard	Carnivores (insectivores)	4	1 (25.0)
African fat-tailed gecko	Carnivores	9	4 (44.45)
Crested gecko	Carnivores (insectivores)	4	2 (50.0)
Leopard gecko	Carnivores (insectivores)	1	1 (100)
Tokay gecko	Carnivores	2	1 (50.0)
Chinese water dragon	Omnivorous	1	1 (100)
Green iguana	Herbivorous	7	2 (28.58)
Veiled chameleon	Carnivores (insectivores)	5	2 (40.0)
Ocelot gecko	Carnivores (insectivores)	3	1 (33.34)
Baja blue rock lizard	Omnivorous	1	1 (100)
Gold tegus	Omnivorous	2	1 (50.0)
Rock monitor	Carnivores	4	2 (50.0)
Central American boa	Carnivores	1	1 (100)
Eastern Kingsnake	Carnivores	2	1 (50.0)
Diadem snake	Carnivores	1	1 (100)
Boid snake	Carnivores	1	1 (100)
Sand boa	Carnivores	2	1 (50.0)
Corn snake	Carnivores	1	1 (100)
Ball python	Carnivores	1	1 (100)
Horsfield tortoise	Herbivorous	2	1 (50.0)
Greek tortoise	Herbivorous	5	1 (20.0)
Hermann’s tortoise	Herbivorous	1	-
Marginated tortoise	Herbivorous	2	-
Chinese pond turtle	Omnivorous	1	1 (100)
Red-eared terrapin	Omnivorous	2	*-*
African helmeted turtle	Omnivorous	2	-
Total		67	29 (43.28)

**Table 4 antibiotics-12-01203-t004:** Serovars of the isolated *Salmonella* from the different animal-keeping places.

Sample Origin	*Salmonella* Subspecies	Serovars
Pet shops	*Salmonella enterica*	Hadar 6.8:z10:e,n,x (n = 1)
		Newport 6.8:e,h:1,2 (n = 2)
		Panama 9.12:l,v:1,5 (n = 1)
		Pomona 28:y:1,7 (n = 1)
		Sandiego 4.12:e,h:e,n,z15 (n = 1)
		Cotham 28:i:1,5 (n = 1)
	*Salmonella houtenae*	16:z4.z32 (n = 1)
		16: z36 (n = 1)
	*Salmonella diarizonae*	42: k: z35 (n = 1)
Private owner	*Salmonella enterica*	Newport 6.8:e,h:1,2 1 (n = 2)
		Lattenkamp 45: z35:1,5 3 (n = 1)
		Paratyphi 4.12: b:1,2 (n = 1)
	*Salmonella arizonae*	44: z4.z23 (n = 3)
	*Salmonella diarizonae*	60:r:e,n,x,z15 (n = 1)
		47:z10:z35 (n = 1)
		50:z52:z35 (n = 2)
	*Salmonella houtenae*	11:z4.z23 (n = 2)

n—number of serovars identified (total serovars, n = 23).

**Table 5 antibiotics-12-01203-t005:** Antimicrobial resistances of the different *Salmonella enterica* strains.

Antimicrobial Categories	Antimicrobial Agents	Number of Isolated *Salmonella* (%)		
		S	I	R
Aminoglycosides	Amikacin (AN)	20/32; 62.5%	3/32; 9.37%	9/32; 28.12%
	Gentamicin (GM)	4/32; 12.55%	1/32; 3.12%	27/32; 84.37%
	Tobramycin I	10/32; 31.25%	4/32; 12.55%	18/32; 56.25%
Penicillin	Ampicillin (AM)	22/32; 68.75%	3/32; 9.37%	7/32; 21.87%
Penicillin with beta lactamase inhibitor	Piperacillin/tazobactam (TZP)	32/32; 100%	0/32	0/32
	Ampicillin/sulbactam (SAM)	32/32; 100%	0/32	0/32
First-generation cephalosporin	Cefazolin (CZ)	32/32; 100%	0/32	0/32
Third-generation cephalosporin	Ceftazidime (CAZ)	32/32; 100%	0/32	0/32
	Ceftriaxone (CRO)	26/32; 81.25%	4/32; 12.55%	2/32; 6.25%
Fourth-generation cephalosporin	Cefepime (FEP)	32/32; 100%	0/32	0/32
Fluoroquinolones	Ciprofloxacin (CIP)	25/32; 78.12%	3/32; 9.37%	4/32; 12.5%
	Levofloxacin (LEV)	32/32; 100%	0/32	0/32
Carbapenem agents	Ertapenem (ETP)	30/32; 93.75%	0/32	2/32; 6.25%
	Imipenem (IPM)	30/32; 93.75%	0/32	2/32; 6.25%
Nitrofuran derivative	Nitrofurantoin (FT)	18/32; 56.25%	3/32; 9.37%	11/32; 34.38%
Diaminopyrimidine with sulfonamide	Trimethoprim/Sulfamethoxazole (SXT)	5/32; 15.62%	4/32; 12.55%	23/32; 71.87%

Clinical and Laboratory Standards Institute (CLSI). Performance Standards for Antimicrobial Susceptibility Testing 28th ed. CLSI supplement M100, 2018. Total *Salmonella enterica* strains tested, n = 32.

**Table 6 antibiotics-12-01203-t006:** Antimicrobial resistance patterns of *Salmonella* subspecies and serovars.

*Salmonella* Subspecies	Serovars (n = 23)	AMC Patterns	Number of Antibiotics/MDR (Yes or Not)
*Salmonella enterica*	Hadar 6.8.:z10:e,n,x (n = 1)	GM-SXT-TM-FT	4/yes
	Newport 6.8:e,h:1,2 (n = 2)	GM-SXT	2/not
		GM-SXT-FT	3/yes
	Panama 9.12:l,v:1,5 (n = 1)	GM-TM	2/not
	Pomona 28:y:1,7 (n = 1)	GM-SXT-FT	3/yes
	Sandiego 4.12:e,h:e,n,z15 (n = 1)	GM-SXT-CIP	3/yes
	Cotham 28:i:1,5 (n = 1)	GM-SXT	2/not
*Salmonella houtenae*	16:z4.z32 (n = 1)	TM-FT-CRO	3/yes
	16:z36 (n = 1)	GM-SXT-TM	3/yes
*Salmonella diarizonae*	42:k: z35 (n = 1)	GM-SXT-TM-FT	4/yes
*Salmonella enterica*	Newport 6.8:e,h:1,2 1 (n = 2)	GM-SXT	2/not
		GM-SXT-FT	3/yes
	Lattenkamp 45:z35:1,5 3 (n = 1)	GM-SXT	2/not
	Paratyphi 4.12:b:1,2 (n = 1)	GM-SXT-IPM	3/yes
*Salmonella arizonae*	44:z4.z23 (n = 3)	GM-TM-FT	3/yes
		GM-SXT	2/not
		GM-TM-ETP	3/yes
*Salmonella diarizonae*	60:r:e,n,x,z15 (n = 1)	SXT-TM	2/not
	47:z10:z35 (n = 1)	GM-SXT-CIP	3/yes
	50:z52:z35 (n = 2)	GM-TM-FT	3/yes
		GM-SXT	2/not
*Salmonella houtenae*	11:z4.z23 (n = 2)	GM-TM	2/not
		GM-SXT-TM	3/yes

n = number of samples; total strains serotyped: 23.

**Table 7 antibiotics-12-01203-t007:** The distribution of reptiles examined according to the species and type of diet.

Order	Family	Species	Common Name	Type of Feeding
*Squamata* *(lizards)*	*Gerrhosauridae*	*Zonosaurus laticaudatuis*	Western girdled lizard	Carnivores (insectivores)
	*Eublepharidae*	*Hemitheconyx caudicinctus*	African fat-tailed gecko	Carnivores
	*Diplodactylidae*	*Correlophus ciliatus*	Crested gecko	Carnivores (insectivores)
	*Eublepharidae*	*Eublepharis macularius*	Leopard gecko	Carnivores (insectivores)
	*Gekkonidae*	*Gecko gecko*	Tokay gecko	Carnivores
	*Agamidae*	*Physignathus cocincinus*	Chinese water dragon	Omnivorous
	*Iguanidae*	*Iguana iguana*	Green iguana	Herbivorous
	*Chamaeleonidae*	*Chamaleo calyptratus*	Veiled chameleon	Carnivores (insectivores)
	*Gekkonidae*	*Paroedura picta*	Ocelot gecko	Carnivores (insectivores)
	*Phrynosomatidae*	*Petrosaurus thalassinus*	Baja blue rock lizard	Omnivorous
	*Tupinambinae*	*Tupinambis teguixin*	Gold tegus	Omnivorous
	*Varanidae*	*Varanus albigularis*	Rock monitor	Carnivores
*Serpentes*	*Boidae*	*Boa constrictor imperator*	Central American boa	Carnivores
*Squamata* *(snakes)*	*Colubridae*	*Lampropeltis getula*	Eastern Kingsnake	Carnivores
	*Colubridae*	*Spalerosophis diadema*	Diadem snake	Carnivores
	*Sanziniidae*	*Acrantophis madagascariensis*	Boid snake	Carnivores
	*Boidae*	*Gongylophis colubrinus*	Sand boa	Carnivores
	*Colubridae*	*Elaphe guttata*	Corn snake	Carnivores
	*Pithonidae*	*Python regius*	Ball python	Carnivores
*Chelonia*	*Testudinidae*	*Testudo horsfieldii*	Horsfield tortoise	Herbivorous
	*Testudinidae*	*Testudo graeca*	Greek tortoise	Herbivorous
	*Testudinidae*	*Testudo hermanni*	Hermann’s tortoise	Herbivorous
	Testudinidae	*Testudo marginata*	Marginated tortoise	Herbivorous
	*Testudinidae*	*Mauremys reevesii*	Chinese pond turtle	Omnivorous
	*Eminidae*	*Trachemys scripta elegans*	Red-eared terrapin	Omnivorous
*Testudines*	*Pelomedusidae*	*Pelomedusa subrufa*	African helmeted turtle	Omnivorous

## Data Availability

Not applicable.

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
