# Peer review of "Surveys on Pet-Reptile-Associated Multi-Drug-Resistant Salmonella spp. in the Timișoara Metropolitan Region—Western Romania"

_antibiotics, 2023, doi:10.3390/antibiotics12071203_

Round 1
Reviewer 1 Report
The manuscript titled "Surveys on pet reptiles-associated multi-drug resistance Salmonella spp in the Timișoara Metropolitan Region-Western Romania: Implication for Public health" presents valuable research. However, it needs extensive modifications before acceptance for publication.
Firstly, the authors consistently discuss multi-drug resistance (MDR) Salmonella spp throughout the manuscript, but in the methodology section, they do not explain how the isolates were classified as MDR. It is important to clarify the criteria used for determining MDR.
Secondly, there are several grammatical and syntax mistakes, such as the sentence "The highest percentages of AMR were found in GM." It would be helpful to revise these errors for improved clarity and readability.
Additionally, the authors mention that "reptiles from private owners are exposed to better hygiene practices and less stressful environments, leading to lower Salmonella shedding." It would be beneficial to elaborate on the link between stress and Salmonella acquisition or shedding to provide a clearer understanding of the connection.
Furthermore, the authors state that "cloacal swabs collected were more sensitive to Salmonella isolation than other samples, such as skin or oral cavity." It is essential to clarify what is meant by "sensitive to" in this context.
The manuscript contains several mistakes that are changing the intended meaning.
The quality of English language is quite poor and manuscript contains several mistakes that are changing the intended meaning.
Author Response
Answers for reviewer 1
Dear reviewer, thanks for taking the time to review our manuscript and for your close attention to detail. We highly appreciate your corrections, suggestions, and comments, which significantly improved the quality of the submission. During the revision process, we tried to do our best to address each of these successfully. Please see below our responses in a point-by-point fashion to the raised concerns.
To be easily findable in the revised manuscript we marked all our answers/corrections in green.
With due respect for your hard work and expertise,
Lecturer DVM Ph.D., M.Sc. DEGI Janos
The manuscript titled "Surveys on pet reptiles-associated multi-drug resistance Salmonella spp in the Timișoara Metropolitan Region-Western Romania: Implication for Public health" presents valuable research. However, it needs extensive modifications before acceptance for publication.
Q1: Firstly, the authors consistently discuss multi-drug resistance (MDR) Salmonella spp throughout the manuscript, but in the methodology section, they do not explain how the isolates were classified as MDR. It is important to clarify the criteria used for determining MDR.
Answer: With respect, but this aspect is mentioned in the manuscript: The isolates resistant to three or more antimicrobials were classified as multidrug-resistant (L328-329).
Q2: Secondly, there are several grammatical and syntax mistakes, such as the sentence "The highest percentages of AMR were found in GM." It would be helpful to revise these errors for improved clarity and readability. L117
Answer: We corrected in the revised manuscript, by reformulating the paragraph (L133-137).
Q3: Additionally, the authors mention that "reptiles from private owners are exposed to better hygiene practices and less stressful environments, leading to lower Salmonella shedding." It would be beneficial to elaborate on the link between stress and Salmonella acquisition or shedding to provide a clearer understanding of the connection. L144-146
Answer: In support of what was stated, the following bibliographic sources was cited:
Marin C, Lorenzo-Rebenaque L, Laso O, Villora-Gonzalez J, Vega S. Pet Reptiles: A Potential Source of Transmission of Multidrug-Resistant Salmonella. Front Vet Sci. 2021 Jan 6;7:613718. doi: 10.3389/fvets.2020.613718. PMID: 33490138; PMCID: PMC7815585.
Marin C, Vega S, Marco-Jiménez F. Tiny turtles purchased at pet stores are a potential high risk for Salmonella human infection in the Valencian Region, Eastern Spain. Vector-Borne Zoonotic Dis. (2016) 16:455–60. doi: 10.1089/vbz.2016.1950.
Q4: Furthermore, the authors state that "cloacal swabs collected were more sensitive to Salmonella isolation than other samples, such as skin or oral cavity." It is essential to clarify what is meant by "sensitive to" in this context. L148-149
Answer: In support of what was stated, the following bibliographic source was cited:
Marin C, Lorenzo-Rebenaque L, Laso O, Villora-Gonzalez J, Vega S. Pet Reptiles: A Potential Source of Transmission of Multidrug-Resistant Salmonella. Front Vet Sci. 2021 Jan 6;7:613718. doi: 10.3389/fvets.2020.613718. PMID: 33490138; PMCID: PMC7815585.
Q5: The manuscript contains several mistakes that are changing the intended meaning.
Answer: The mistakes have been corrected and we hope that the revised version is what offers you the possibility of following the description of the study, without understanding anything else. The entire manuscript went through a major revision, precisely to eliminate these mistakes and errors.

Reviewer 2 Report
Dear authors,
in this manuscript you show the prevalence of salmonella in different samples taken from reptiles of different species, with reference to the housing and feeding conditions. For isolated salmonella, species and serotype were determined, as well as the susceptibility to antimicrobials. Through the Results, the relation of the type of sample, the origin of reptiles, conditions of keeping and nutrition with the salmonella species and serovar determined. The shortcomings in this research are related to Introduction (MDR is not mentioned); in Materials and Methods, molecular analyses are mentioned, but the results of the same are not visible in the chapter Results or described through Discussion. Also, the descriptions of the methods used in this study, do not allow repeatability (e.g. part about serotyping and antimicrobial susceptibility testing). The disadvantage could also be seen in the questionnaire filled out by the owners - questions about the health status of the owner, their habits when dealing with reptiles, possible confirmations of the occurrence of salmonellosis could give an insight into the real danger for reptile owners.
Specific comments:
L21-22 same sentence twice; urban spread?
L 26 cages or terrarium?
L 36-37 human multidrug resistant salmonella?
L 44-45 Salmonella are not zoonotic disease (salmonellosis is a disease)
Table 4- MIC breakpoints are later also mentioned in the text; mg of antimicrobial agent are not mentioned
It is a bit confusing how many S. enterica were used for serotyping and for AMR testing - Table 3 mention 23 S. enterica, Table 4 mention 32 and again Table 5, 23?
L 162, 165 - feeder rodent?
L 189 - sentence needs revision
L 204 - title of the table needs revision
L 238 - XLD agar - it is not clear when the samples(bacteria were streaked on that agar?
L268-269 sentence not clear enough
L 271 - all strains?
The quality of English language should be improved
Author Response
Answers for reviewer 2
Dear reviewer, thanks for taking the time to review our manuscript and for your close attention to detail. We highly appreciate your corrections, suggestions, and comments, which significantly improved the quality of the submission. During the revision process, we tried to do our best to address each of these successfully. Please see below our responses in a point-by-point fashion to the raised concerns.
To be easily findable in the revised manuscript we marked all our answers/corrections in green.
With due respect for your hard work and expertise,
Lecturer DVM Ph.D., M.Sc. DEGI Janos
Dear authors,
in this manuscript you show the prevalence of salmonella in different samples taken from reptiles of different species, with reference to the housing and feeding conditions. For isolated salmonella, species and serotype were determined, as well as the susceptibility to antimicrobials. Through the Results, the relation of the type of sample, the origin of reptiles, conditions of keeping and nutrition with the salmonella species and serovar determined.
Q1: The shortcomings in this research are related to Introduction (MDR is not mentioned);
Answer: Thank you for your observation and pertinent recommendation. We have inserted a paragraph in the revised version, in the Introduction chapter: Antibiotic resistance of bacteria to antimicrobials is currently a primary concern in both human and veterinary medicine. For this reason, epidemiological studies in domestic and wild animals should be performed on a regular basis [Merkevičienė 2022]. Resistant pathogens, including Salmonella enterica, should be of particular attention as these bacteria are very well adapted to different hosts, carry different genes encoding for both virulence and antimicrobial resistance and are currently among the most common infectious agents isolated from humans with food-borne infections. Although many countries have strict antimicrobial use restrictions, enforcement is sometimes lax, resulting in the indiscriminate use of antimicrobials. This abuse has aided in the evolution of multidrug-resistant (MDR) bacterial strains [Merkevičienė 2022, Monte 2019]. Salmonella strains that are resistant to antibiotics have been discovered in reptiles all over the world [Marin 2022, Song 2023]. When humans become infected with resistant Salmonella strains, therapy can be challenging, increasing the probability of treatment failure and, in extreme cases, death [Serwecińska 2020, Teklemariam 2023]. As a result, understanding the antibiotic resistance patterns of Salmonella spp. infections in reptiles is critical (L73-83).
Bibliographic sources added:
- Merkevičienė, L.; Butrimaitė-Ambrozevičienė, Č.; Paškevičius, G.; Pikūnienė, A.; Virgailis, M.; Dailidavičienė, J.; Daukšienė, A.; Šiugždinienė, R.; Ruzauskas, M. Serological Variety and Antimicrobial Resistance in SalmonellaIsolated from Reptiles. Biology 2022, 11, 836. https://doi.org/10.3390/biology11060836
- Monte, D.F.; Lincopan, N.; Fedorka-Cray, P.J.; Landgraf, M., Current insights on high priority antibiotic-resistant Salmonella enterica in food and foodstuffs: a review, Current Opinion in Food Science, 2019, 35-46.
- Marin C, Martín-Maldonado B, Cerdà-Cuéllar M, et al. Antimicrobial Resistant Salmonella in Chelonians: Assessing Its Potential Risk in Zoological Institutions in Spain. Vet Sci. 2022;9(6):264.
- Song, D.; He, X.; Chi, Y.; Zhang, Z.; Shuai, J.; Wang, H.; Li, Q.; Du, M. Cytotoxicity and Antimicrobial Resistance of Salmonella entericaSubspecies Isolated from Raised Reptiles in Beijing, China. Animals 2023, 13, 315.
- Serwecińska, L. Antimicrobials and Antibiotic-Resistant Bacteria: A Risk to the Environment and to Public Health. Water2020, 12, 3313.
- Teklemariam, A.D.; Al-Hindi, R.R.; Albiheyri, R.S.; Alharbi, M.G.; Alghamdi, M.A.; Filimban, A.A.R.; Al Mutiri, A.S.; Al-Alyani, A.M.; Alseghayer, M.S.; Almaneea, A.M.; et al. Human Salmonellosis: A Continuous Global Threat in the Farm-to-Fork Food Safety Continuum. Foods 2023, 12, 1756
Q2: in Materials and Methods, molecular analyses are mentioned, but the results of the same are not visible in the chapter Results or described through Discussion.
Answer: we corrected in the revised manuscript: We entered the recommended information in the Results chapter (L88-89).
Q3: Also, the descriptions of the methods used in this study, do not allow repeatability (e.g. part about serotyping and antimicrobial susceptibility testing).
Answer: We respect your opinion, but the methods used are those described in a previous work published in Antibiotics by the research group I led, when they were isolated and serotyped, respectively tested for sensitivity to antibiotics in Salmonella strains originating from cats.
Dégi J, Imre K, Herman V, et al. Antimicrobial Drug-Resistant Salmonella in Urban Cats: Is There an Actual Risk to Public Health?. Antibiotics (Basel). 2021;10(11):1404.
Serotyping of the Salmonella isolates was achieved in a pure culture based on the evidence of somatic (O) and flagellar (H) antigens through reactions with specific antisera, according to the manufacturer's recommendations (L306-308).
Antimicrobial susceptibility testing of the isolated Salmonella strains was achieved with the Vitek 2® automated equipment and the AST GN67 card and the system automatically processed the obtained results (L315-327).
Q4: The disadvantage could also be seen in the questionnaire filled out by the owners - questions about the health status of the owner, their habits when dealing with reptiles, possible confirmations of the occurrence of salmonellosis could give an insight into the real danger for reptile owners.
Answer: We appreciate pertinent comments on the topic under discussion. Such questions were not included, because they are the subject of ongoing research, to establish the correlation between the presence of MDR Salmonella strains in reptiles and the different habits regarding the feeding, handling, storage of food intended for reptiles and different professional categories exposed to the risk of disease with these zoonotic pathogens. The results obtained in the current study indicate that there is an important risk for public health due to the presence of MDR Salmonella enterica strains.
Specific comments:
Q5: L21-22 same sentence twice; urban spread?
Answer: Thank you for the pertinent observation. I removed the phrase that was mentioned twice in the manuscript. To avoid confusion, we changed the word spread to expansion (L21-22).
Q6: L 26 cages or terrarium?
Answer: we corrected with terrarium
Q7: L 36-37 human multidrug resistant salmonella?
Answer: we corrected with reptile’s multi-drug resistant Salmonella (L33)
Q8: L 44-45 Salmonella are not zoonotic disease (salmonellosis is a disease)
Answer: we corrected with zoonotic pathogens (L44).
Q9: Table 4- MIC breakpoints are later also mentioned in the text; mg of antimicrobial agent are not mentioned
Answer: The table has been revised and the data related to MIC breakpoints have been deleted and are presented in a table with additional data, in order to offer those who read the work a better visualization and understanding of the data presented (table 5, L138).
Q10: It is a bit confusing how many S. enterica were used for serotyping and for AMR testing - Table 3 mention 23 S. enterica, Table 4 mention 32 and again Table 5, 23?
Answer: 23 Salmonella strains (L108-109), out of the 29 selected, were subjected to the serological typing test. 6 strains were not viable and were excluded from the study. These aspects are presented in table 4 (L124-129). All these 23 serotyped strains were tested for sensitivity to antibiotics and the resistance patterns are shown in table 6 (L146).
Table 5 (L138) shows the results obtained from the antibiotic resistance testing of 32 strains of Salmonella enterica, out of the 41 isolated in our study. Of the 41 strains, 9 strains were not viable and were excluded from this evaluation.
Q11: L 162, 165 - feeder rodent?
Answer: Yes, it is about feeder rodents, which can be mice and rats—both frozen and live—used to feed some reptiles, such as certain snakes and lizards. Feeder rodents go by various names, depending on their age: pinkies (1 to 5 days old), fuzzies (6 to 13 days old), hoppers (14 to 20 days old), and adults (21 days and older).
The FSA (Food and Standards Agency) is urging reptile owners who purchase certain feeder rodents for their pets to take extra precautions and always wash their hands to avoid becoming ill with salmonellosis.
Epidemiological investigations and whole genome sequencing have again confirmed the link between a Salmonella outbreak in people who have become unwell and feeder rodents used to feed reptiles and some other animals distributed in the UK by this specific importer.
Q12: L 189 - sentence needs revision
Answer: Thank you for your observation and pertinent recommendation. We reformulated: All sample techniques were carried out in accordance with national standards and regulations. L 222-223.
Q13: L 204 - title of the table needs revision
Answer: Thank you for your observation and pertinent recommendation. We reformulated the title of the table. The distribution of the number of reptiles examined according to the species and the type of diet. Table 6, from line 204, became after the major revision of the manuscript Table 7, from line 237.
Q14: L 238 - XLD agar - it is not clear when the samples(bacteria were streaked on that agar?
Answer: The inoculation of presumptive strains of Salmonella spp., on XLD agar was done after testing on the McConkey medium (L272-277).
Q15: L268-269 sentence not clear enough
Answer: we corrected in the revised manuscript: Salmonella O and Salmonella H antisera (SSI Diagnostica A/S, Hillerød, Denmark) were used according to the manufacturer's recommendations. L307-308.
Q16: L 271 - all strains?
Answer: we corrected in the revised manuscript: thirty-two out of 41 Salmonella strains isolated were viable after culture were included in the antimicrobial susceptibility study. L312-313.

Reviewer 3 Report
General comment
The title of this manuscript is Surveys on pet reptiles-associated multidrug resistance Salmonella spp in the Timișoara Metropolitan Region-Western Romania: Implication for Public Health. However, no data and analysis support this study's association between pet reptiles and MDR. The author should reconsider the title.
To be more precise and easier to read, I would recommend revising the results following these:
1. Prevalence of salmonella
2. Sample type: oral cavity, skin (normal skin: environment representation, cloacal swab: an animal is a reservoir or gets infected with Salmonella.
3. Risk factors: 4 factors x Salmonella finding. Factors:
a. feed,
b. keeping captive place of reptiles: pet shops and households
c. reptile captive density (alone/ two or more): density of captive effect to reinfect of Salmonella.
4. Antimicrobial susceptibility profiles of isolated Salmonella
5. Multidrug resistance (MRD) patterns
6. Association between MRD finding and pet reptiles with different factors as the author mentions in an introduction line 79. The result of risk factors could be present with p-value and % CI.
Specific comment
Abstract
The abstract is overclaiming the conclusion about the findings of the study. “The findings of the study undertaken by our team reveal that human………can be transmitted by reptiles.” This study just shows the prevalence and MRD but does not prove the transfer of Salmonella from reptile to human. It could be written in discussion or suggestion in conclusion part.
Introduction
Line 58-59 please move this sentence to the previous paragraph.
Line 72-76 It would fit in the discussion part.
Results
Why did the authors present a positive percentage in mean and SE (25.46±2.3)? How did calculate it?
The p-value in the result part should be present in the table and please explain which statistical analysis did apply.
Table 2, as the authors mention in text line 104-106 about the first to third category. Please indicate them in table 2.
Table 3, I would suggest revising the name of table to “Serovars of the isolated Salmonella from the different animal keeping places.” And move (n=23) under table as a remark.
Line 115-121 please summarize in percentage based on type of antibiotic for example beta-lactam...., fluoroquinolone........ It is easier to see MDR.
Table 4, 1. please move interpretation categories to supplement part. It is unnecessary to be show in result. 2. add the number of isolates in each category (S I and R). And 3. remove last column.
Table 5 is not an antimicrobial pattern. The author should revise the table presenting for example this below.
|
Serovars |
AMC pattern |
Number of isolate (%) |
|
Salmonella Hadar |
AN-CIP-GM-FT-TM-SXT |
1 |
Discussion
The discussion part could be revise following the objectives of this study as the multi-drug resistance Salmonella and potential risk factors in pet stores and households.
Material and Methods
4.1 sample collection:
· Line 183-185 "Salmonella identification was based on ISO 6579-1:2017 (Annex D), serotyped by the Kauffman-White-Le-Minor technique, and antibiotic susceptibility was assessed according to Decision 2013/652." Move to 4.2 bacterial isolation.
· Table 6 presents only species of reptiles in this study. Move the last column (percentage of Salmonella finding samples) to the result part.
· Describe the retile based on their feed, for example, "Carnivore, herbivore, or both" because their feed source can be contaminated Salmonella.
· Diet classification in this study: live prey, fresh meat (what kinds of meat feed to reptiles in this study? The different meat types can be different risks of salmonella contamination), or frozen meat (the freezing and thawing method affects bacterial contamination on meat).
· What does it mean of "processed feed"? Heat cooking or feed pellet process of any process. This process can affect a load of salmonella contamination, and the authors could discuss it as a risk factor. Please describe it for the reader.
4.2 Minor revision Refer to comment on 4.1.
4.3 please change "Molecular analyses" to "Molecular detection of invA gene." It is more precise to the objective of molecular analysis.
4.4 Please cite the original paper: Grimont P.A., Weill F.X. Antigenic formulae of the Salmonella serovars. WHO Collab. Cent. Ref. Res. Salmonella. 2007; 9:1–166.
And
How many isolates of Salmonella per positive sample did serotyping? Please describe it in the text.
4.5 There are two references to the antimicrobial susceptibility method and interpretation. On table 4, the authors did use CLSI for interpretation, but in 4.5 did apply EUCAST. Please clarify this point.
4.6 Why do the authors present data by mean and SE? It could present a percentage of findings and analyze the potential risk factors. These would be more attractive outcomes of this study.

This manuscript is understandable in English.
Author Response
Answers for reviewer 3
Dear reviewer, thanks for taking the time to review our manuscript and for your close attention to detail. We highly appreciate your corrections, suggestions, and comments, which significantly improved the quality of the submission. During the revision process, we tried to do our best to address each of these successfully. Please see below our responses in a point-by-point fashion to the raised concerns.
To be easily findable in the revised manuscript we marked all our answers/corrections in green.
With due respect for your hard work and expertise,
Lecturer DVM Ph.D., M.Sc. DEGI Janos
The title of this manuscript is Surveys on pet reptiles-associated multidrug resistance Salmonella spp in the Timișoara Metropolitan Region-Western Romania: Implication for Public Health. However, no data and analysis support this study's association between pet reptiles and MDR. The author should reconsider the title.
Answer: Thank you for the relevant suggestion and I removed the part of implication for public health from the title of the revised manuscript (L4).
Q1: To be more precise and easier to read, I would recommend revising the results following these:
- Prevalence of salmonella
- Sample type: oral cavity, skin (normal skin: environment representation, cloacal swab: an animal is a reservoir or gets infected with Salmonella.
- Risk factors: 4 factors x Salmonella finding. Factors:
- feed,
- keeping captive place of reptiles: pet shops and households
- reptile captive density (alone/ two or more): density of captive effect to reinfect of Salmonella.
- Antimicrobial susceptibility profiles of isolated Salmonella
- Multidrug resistance (MRD) patterns
- Association between MRD finding and pet reptiles with different factors as the author mentions in an introduction line 79. The result of risk factors could be present with p-value and % CI.
Answer: Thank you for the recommendation and I have presented the data respecting the requirements described previously, in the results and discussions chapter.
Specific comment
Q2: Abstract
The abstract is overclaiming the conclusion about the findings of the study. “The findings of the study undertaken by our team reveal that human………can be transmitted by reptiles.” This study just shows the prevalence and MRD but does not prove the transfer of Salmonella from reptile to human. It could be written in discussion or suggestion in conclusion part.
Answer: we moved the paragraph in the conclusion part, in the revised manuscript (L343-347).
Q3: Introduction
Line 58-59 please move this sentence to the previous paragraph.
Answer: I respect your opinion and I ask you which previous paragraph, in the abstract chapter?
Line 72-76 It would fit.
Answer: With respect to your opinion, I believe that the information provided in this paragraph contains general elements about the risks regarding Salmonella infection and about the pathogenesis. These details do not help highlight the results obtained in our study and for this reason we did not move to another chapter.
Q4: Results
Why did the authors present a positive percentage in mean and SE (25.46±2.3)? How did calculate it?
Answer: The standard error (SE) of a statistic is the approximate standard deviation of a statistical sample population. The standard error is a statistical term that measures the accuracy with which a sample distribution represents a population by using standard deviation. In statistics, a sample mean deviates from the actual mean of a population; this deviation is the standard error of the mean.
We calculated the standard deviation of the sample and the population, to assess the risk. Perhaps for these reasons, the data presented were not clear enough and difficult to follow. Analyses were performed using commercially available software (SPSS software package—version 28.0.1.1, Chicago, USA).
The p-value in the result part should be present in the table and please explain which statistical analysis did apply.
Answer: The results were analysed using basic descriptive statistics. The 95 per cent confidence intervals (CI) for the Salmonella enterica strains isolated from reptiles, were calculated, and estimated to compare between different factors. The obtained results were statistically interpreted using the SPSS statistical analysis software package, version 28.0.1.1, Chicago, USA. A nonparametric Pearson’s chi-squared (χ2) test was used to find any possible associations between the Salmonella infection status and the recorded epidemiological data. Differences were established as statistically significant when p-value ≤ 0.05 (L336-341).
Table 2, as the authors mention in text line 104-106 about the first to third category. Please indicate them in table 2.
Answer: Corresponds to the description with what is in the table 2 (L113).
Table 3, I would suggest revising the name of table to “Serovars of the isolated Salmonella from the different animal keeping places.” And move (n=23) under table as a remark.
Answer: we corrected in the revised manuscript and the changes can be found in the in the table 4: Serovars of the isolated Salmonella from the different animal keeping places (L129).
Line 115-121 please summarize in percentage based on type of antibiotic for example beta-lactam...., fluoroquinolone........ It is easier to see MDR.
Answer: we corrected in the revised manuscript and the changes can be found in the in the table 5: Antimicrobial resistance of the different Salmonella enterica strains (L138), respectively L133-137.
Table 4, 1. please move interpretation categories to supplement part. It is unnecessary to be show in result. 2. add the number of isolates in each category (S I and R). And 3. remove last column.
Answer: we corrected in the revised manuscript and the changes can be found in the table 5: Antimicrobial resistance of the different Salmonella enterica strains (L138).
Table 5 is not an antimicrobial pattern. The author should revise the table presenting for example this below.
|
Serovars |
AMC pattern |
Number of isolate (%) |
|
Salmonella Hadar |
AN-CIP-GM-FT-TM-SXT |
1 |
Answer: we corrected in the revised manuscript and the changes can be found in the table 6: Antimicrobial resistance pattern of Salmonella subspecies and serovar (L146).
Q5: Discussion
The discussion part could be revise following the objectives of this study as the multi-drug resistance Salmonella and potential risk factors in pet stores and households.
Answer: we revised this chapter as recommended and we added this paragraph: Antibiotics used to treat infections in humans and pet reptiles were found to have high levels of resistance from some isolated strains of Salmonella in the current study, raising the possibility that humans could become infected with multidrug-resistant Salmonella through contact with reptiles. Fluoroquinolones are an important class of antibiotics used to treat a variety of human and animal (including reptiles) infections, and they are especially effective against salmonellosis. To keep fluoroquinolones as effective as feasible, they must be used with caution, antibiotic residues in food must be checked on a regular basis, and extensive monitoring for the formation of bacterial resistance in both animals and humans must be provided [46,47]. Carbapenems (ertapenem and imipenem), known as "last resort" antibiotics for use in cases where drug resistance monitoring is required, are required to establish any possible links between bacterial reservoirs and to limit the bidirectional transfer of encoding genes between Salmonella spp. and other commensal or pathogenic bacteria. S. enterica is a nosocomial infection in humans, and the frequency with which it is discovered to be resistant to both carbapenems is low when compared to other Enterobacteriaceae [48,49]. (L197-207).
Q6: Material and Methods
4.1 sample collection:
- Line 183-185 "Salmonella identification was based on ISO 6579-1:2017 (Annex D), serotyped by the Kauffman-White-Le-Minor technique, and antibiotic susceptibility was assessed according to Decision 2013/652." Move to 4.2 bacterial isolation.
Answer: we corrected in the revised manuscript, and we moved to subsection 4.2. (L259-261)
- Table 6 presents only species of reptiles in this study. Move the last column (percentage of Salmonella finding samples) to the result part.
Answer: we corrected in the revised manuscript and move the last column in table 3: The distribution of Salmonella strains according to the reptile species and the type of diet (L121).
- Describe the reptile based on their feed, for example, "Carnivore, herbivore, or both" because their feed source can be contaminated Salmonella.
Answer: we corrected in the revised manuscript. Table 7. The distribution of the number of reptiles examined according to the species and the type of diet (L237-238).
- Diet classification in this study: live prey, fresh meat (what kinds of meat feed to reptiles in this study? The different meat types can be different risks of salmonella contamination), or frozen meat (the freezing and thawing method affects bacterial contamination on meat).
Answer: in this study, according to the data provided by the breeders, owners, the following types of food were used: live prey, fresh meat, frozen meat, fruit, vegetables and commercially manufactured reptile food.
- What does it mean of "processed feed"? Heat cooking or feed pellet process of any process. This process can affect a load of salmonella contamination, and the authors could discuss it as a risk factor. Please describe it for the reader.
Answer: it mainly refers to frozen food, which regardless of the defrosting method chosen, can be considered a critical point of contamination and respectively the subsequent storage and handling of this type of food by the owners, etc.
Just like when handling raw human food, there is a risk of Salmonella when handling raw or frozen and defrosted reptile foods, such as rats, mice, and chicks, as freezing does not kill Salmonella bacteria, this applies to all frozen reptile foods, and not just the batches affected by the product recall in question. We recommend wearing gloves any time you handle frozen reptile foods, and to wash your hands thoroughly with soap and water afterwards.
It is important not to allow your frozen reptile food to defrost in the time between purchasing and getting home, and not to defrost and refreeze the frozen feeders at any point. Where possible, we recommend keeping frozen reptile food separate from any human foods, in a separate freezer, however, in cases where this is not possible, the feeders should not be allowed to encounter any human foods that may be in the same freezer. It is also worth double-bagging your frozen reptile food to minimize the risk of cross-contamination.
The manuscript describes these aspects regarding the food categories for reptiles and what it includes.: The diet was classified as food of animal origin (including live prey, fresh meat, and frozen meat), food of vegetable origin (including fruit and vegetables) and processed (including commercially manufactured reptile food) (L245-247).
4.2 Minor revision Refer to comment on 4.1.
4.3 please change "Molecular analyses" to "Molecular detection of invA gene." It is more precise to the objective of molecular analysis.
Answer: we corrected in the revised manuscript, according to the recommendations (L283).
4.4 Please cite the original paper: Grimont P.A., Weill F.X. Antigenic formulae of the Salmonella serovars. WHO Collab. Cent. Ref. Res. Salmonella. 2007; 9:1–166.
Answer: we added this work to the bibliography list in the revised manuscript
And
How many isolates of Salmonella per positive sample did serotyping? Please describe it in the text.
Answer: From the 41 isolated strains of Salmonella from reptiles, 29 strains selected for serotyping and 23 were viable after culture and were serotyped (L124-125)
4.5 There are two references to the antimicrobial susceptibility method and interpretation. On table 4, the authors did use CLSI for interpretation, but in 4.5 did apply EUCAST. Please clarify this point.
Answer: Due to a regrettable error, EUCAST was passed. I corrected in the revised manuscript, and I modified this aspect at the bibliography level as well. Antimicrobial susceptibility was tested according to the Clinical and Laboratory Standards Institute (CLSI) guidelines (L312-314).
4.6 Why do the authors present data by mean and SE? It could present a percentage of findings and analyze the potential risk factors. These would be more attractive outcomes of this study.
Answer: we corrected in the revised manuscript. The result was analyzed using basic descriptive statistics. The 95 per cent confidence intervals (CI) for Salmonella enterica prevalence in reptiles were calculated and compare the incidence of Salmonella between different factors.
Q7: This manuscript is understandable in English.
Answer: The revised manuscript was sent to the English language editing service and corrected, so that it corresponds to the requirements and standards of the English language. I am attaching the certificate of editing in English.

Round 2
Reviewer 1 Report
The authors have improved the manuscript and can be accepted
The English language is improved now
Author Response
Answers for the reviewer 1 – R2
We thank the referee for the careful and insightful review of our manuscript.

Reviewer 2 Report
Dear authors,
thank you for taking into consideration all the comments given by the reviewer.
Best regards
Minor changes still needed
Author Response
Answers for the reviewer 2 – R2
We would like to thank the reviewer for their thoughtful comments and efforts towards improving our manuscript.

Reviewer 3 Report
the recommendation for minor revision.

Author Response
Answers for the reviewer 3 R2
Minor recommend.
Q1: 1. Abstract: I would recommend removing this sentence …,and can be transmitted to humans at line no. 34 due to your finding in this study.
Answer: we removed the sentence in revised manuscript
Q2: 2. Table 2: I recommend minor revision, for example, as below.
Answer: we modified the table as required
Q3: Line 112: Salmonella carriage and diets proved to be tightly associated (p = 0.4809; 95% CI=-112 21.3475% to 38.9895%). Please reconsider this sentence because of this p-value.
Answer: we corrected this aspect and mention in the revised manuscript: …. Salmonella carriage and diets were not found to be tightly associated (p = 0.4809; 95% CI=-21.3475% to 38.9895%),... or there to be a correlation between the value of p and the description of the results obtained.
Q4: Line 112-116: The authors cite Table 2 in a text about the first, second, and third categories, but these categories did not mention and related to the table. It would precisely understand readers that the authors revise/write something in the text related to your table.
Answer: we modified the table as required and I introduced the three categories of diet in the table, so that there is a correlation between the description and the content of the table.
Q5: Table 5 I would recommend revising to
Table 5. Antimicrobial resistances of the different Salmonella enterica strains.
|
|
Antimicrobial categories |
Antimicrobial agents |
Number of isolated Salmonella (%) |
||
|
|
S |
I |
R |
||
Answer: we modified the table as required
Q6: Table 7, please remove “the number” from the table topic “The distribution of the number of reptiles examined according to the species and type of diet.” Because the authors do not present any number in text of table.
Answer: We removed this sentence in the revised manuscript
